# Anti-Inflammatory and Analgesic Properties of the Cannabis Terpene Myrcene in Rat Adjuvant Monoarthritis

**DOI:** 10.3390/ijms23147891

**Published:** 2022-07-17

**Authors:** Jason J. McDougall, Meagan K. McKenna

**Affiliations:** Departments of Pharmacology and Anaesthesia, Pain Management & Perioperative Medicine, Dalhousie University, 5850 College Street, Halifax, NS B3H 4R2, Canada; mg398954@dal.ca

**Keywords:** arthritis, cannabis, entourage effect, inflammation, joint damage, pain, terpenes

## Abstract

*Cannabis*-based terpenes are believed to modulate physiological responses to disease and alter the efficacy of cannabinoids in the so-called “entourage effect”. The monoterpene myrcene can reduce nociception produced by noxious thermal and mechanical stimuli as well as reducing acute inflammation. The current study examined the role of myrcene and cannabidiol (CBD) in controlling chronic joint inflammation and pain. Chronic arthritis was induced in male Wistar rats by intra-articular injection of Freund’s complete adjuvant into the right knee. On days 7 and 21 after arthritis induction, joint pain (von Frey hair algesiometry), inflammation (intravital microscopy, laser speckle contrast analysis) and joint histopathology were assessed. Local application of myrcene (1 and 5 mg/kg s.c.) reduced joint pain and inflammation via a cannabinoid receptor mechanism. The combination of myrcene and CBD (200 μg) was not significantly different from myrcene alone. Repeated myrcene treatment had no effect on joint damage or inflammatory cytokine production. These data suggest that topical myrcene has the potential to reduce chronic arthritis pain and inflammation; however, it has no synergistic effect with CBD.

## 1. Introduction

Plants of the genus *Cannabis* have a complex composition consisting of 125 cannabinoids and over 400 other non-cannabinoid chemicals including flavonoids, terpenes, and alkaloids [1]. The terpenes provide a distinct flavour and odour to the different strains of *Cannabis*; however, emerging evidence indicates that these compounds can also exert multiple physiological responses. Myrcene (7-Methyl-3-methylene-1,6-octadiene) is an acyclic monoterpene that exists in two distinct isotopes, alpha and beta. Beta-myrcene (heretofore referred to simply as myrcene) is a colourless liquid with a woody, turpentine odour. In addition to being found in hops, thyme, and bay leaves, myrcene is the most prevalent monoterpene in *Cannabis*. In preclinical studies, administration of essential oils rich in myrcene have been found to have analgesic and anti-inflammatory properties [2,3]. *Eremanthus erythropappus* (10% myrcene) and *Bougainvillea glabra* (4% myrcene) increased hindpaw withdrawal time in mice and rats in response to noxious heat while reducing paw licking in mice injected with formalin [4,5]. In addition, intra-articular injection of an essential oil from *Chamaecyparis obtusa* (26% myrcene) improved weight bearing in the carrageenan model of acute synovitis to a level that was comparable to indomethacin [6]. Intraperitoneal injection of pure myrcene dose-dependently increased paw withdrawal latency in the hot plate test of thermonociception and this effect involved secondary release of endogenous opioids [7]. Repeated oral administration of myrcene has been shown to produce prolonged analgesia in rats treated with intra-plantar prostaglandin E2 whilst avoiding the development of tolerance [8]. Notably, high-dose myrcene had no effect on rotarod activity, indicating that myrcene produces a true anti-nociceptive effect and not merely an impairment in locomotor control [9].

Plant-derived essential oils containing around 10% myrcene have also been found to have anti-inflammatory properties. Oil derived from the plant *Eremanthus erythropappus* was able to reduce oedema and leukocyte extravasation in response to carrageenan in a number of tissues including the hindpaw and lung [4,5]. Another myrcene-rich essential oil reduced the production of inflammatory cytokines and cyclooxygenase-2 (COX-2) within eight hours in a rat model of acute synovitis [6]. Treatment of arthritic human chondrocytes with pure myrcene reduced inducible nitric oxide synthase activity and interrupted the interleukin-1β signaling pathway [10]. Taken together, these data suggest that myrcene and myrcene-containing essential oils have the potential to ameliorate pain and inflammation. Whether these desirable effects of myrcene are applicable to chronic inflammatory joint disease has not been explored.

The cannabinoid receptors CB_1_ and CB_2_ have been located on sensory nerve terminals innervating synovial joints where their activation reduces nociceptor firing and pain [11,12,13]. Cannabinoid receptor-mediated analgesia is achieved by a retrograde signaling process resulting in inhibition of pro-algesic neurotransmitter release from nociceptor terminals [14]. Both cannabinoid receptors have also been located on immunocytes where they can induce both pro- and anti-inflammatory responses [15,16]. In joints, cannabinoid receptor activation locally can alter synovial blood flow, leukocyte trafficking, and joint oedema [17,18,19,20]. Since some terpenes can act directly on cannabinoid receptors and alter the pharmacokinetics of phytocannabinoids [3,21], the interaction between terpenes and the endocannabinoid system warrants further investigation.

Decarboxylation of plant-based cannabidiolic acid results in the formation of the non-euphoric cannabinoid cannabidiol (CBD), which has anti-inflammatory and analgesic properties [19]. Articular administration of CBD reduced the nociceptor firing rate, evoked pain behavior, synovial hyperaemia, and leukocyte trafficking in a model of osteoarthritis [22]. Joint inflammation and thermonociception were similarly attenuated in adjuvant monoarthritic knees and resulted in a decreased expression of inflammatory biomarkers [23]. Systemic CBD was also able to reduce the number of circulating neutrophils and inhibit inflammatory cytokine release in lipopolysaccharide-treated mice, providing further evidence for its anti-inflammatory action [24]. While the beneficial potential for CBD has been repeatedly shown in animal models of arthritis, human data are conflicting, suggesting the need for closer inspection [25,26].

Whole plant extracts from *Cannabis* are known to be more efficacious when given together than the individual isolates administered alone, suggesting a functional synergy between multiple chemical components in the plant [27,28]. One of the first descriptions of this so-called “entourage effect” was proffered by Karniol and Carlini who discovered that CBD potentiated the analgesic capacity of delta-9-tetrahydrocannabinol (Δ^9^-THC) in the hot plate test [29]. Cannabis-based terpenes were found to have cannabinomimetic properties in the tail flick test and enhanced the anti-nociceptive effects of the non-selective cannabinoid agonist WIN55,212-2 [30]. Conversely, potassium ion channel opening by Δ^9^-THC was unaffected by *Cannabis* terpenes, suggesting a lack of entourage effect in cellular hyperpolarization [31]. Hence, a comprehensive investigation into possible entourage effects between cannabinoids and non-cannabinoid molecules in *Cannabis* is warranted.

The objective of this study was to characterize the effect of myrcene on joint pain and inflammation using a model of chronic arthritis. Potential interactions between myrcene and low-dose CBD on joint pathophysiology were also examined.

## 2. Results

### 2.1. Acute Effects of Myrcene on Joint Pain and Inflammation

Local administration of myrcene to adjuvant monoarthritic knees caused a dose-dependent reduction in secondary allodynia over the 3 h time-course (*p* < 0.05 two factor RMANOVA; *n* = 8 animals/group). The greatest effect of myrcene occurred at the 120 min timepoint where the 1 mg/kg dose improved nociception by 211.0 ± 17.93% and the 5 mg/kg dose 269.3 ± 63.27% (Figure 1A). Focusing on the 120 min timepoint (Figure 1B), the analgesic effect of myrcene was blocked by pre-treatment with either the CB_1_-receptor antagonist AM281 (*p* < 0.001 one factor RMANOVA with Bonferroni’s post hoc test; *n* = 8 animals/group) or the CB_2_-receptor antagonist AM630 (*p* < 0.0001). Neither antagonist alone had any effect on joint pain.

In day 7 arthritic knees, local administration of myrcene reduced leukocyte rolling with the maximal effect occurring at 60 min after topical application (*p* < 0.0001; *n* = 6–7 animals/group. Figure 2A). This anti-inflammatory effect was not dose-dependent. Myrcene had no effect on leukocyte adherence or joint blood flow (data not shown). Blockade of CB_2_ receptors with AM630 attenuated the reduction in leukocyte rolling by myrcene (*p* < 0.05); however, AM281 did not affect the anti-inflammatory effects of myrcene (Figure 2B). Neither antagonist alone had any effect on joint inflammation.

### 2.2. Combination Effects of Myrcene and CBD

In these experiments, a low dose of CBD, which in itself had no effect on pain and inflammation, was chosen to see if it could synergize with myrcene to augment antinociception and anti-inflammation. While myrcene alone reduced secondary allodynia and leukocyte rolling, the addition of CBD had no additional effect on these parameters (*p* > 0.05, Figure 3A,B).

### 2.3. Effect of Chronic Myrcene on Hindlimb Pain and Inflammation

Compared to vehicle, repeated administration of myrcene increased the hindpaw withdrawal threshold over 21 days of adjuvant-induced arthritis (*p* < 0.0001; *n*= 7–8; Figure 4). These measurements were carried out prior to myrcene administration on each day so cannot be attributed to any acute effects of the compound.

Chronic myrcene treatment had a wider effect on leukocyte trafficking and joint perfusion than acute administration (Figure 5). In addition to an effect on leukocyte rolling (*p* < 0.01), repeated topical myrcene was able to reduce leukocyte adherence (*p* < 0.05) and synovial blood flow (*p* < 0.01). In day 21 adjuvant monoarthritic knees, topical myrcene reduced leukocyte rolling from 72.6 ± 4.59 leukocytes/min to 47.9 ± 5.02 leukocytes/min; leukocyte adherence from 5 ± 0.8 to 3 ± 0.5 leukocytes/100 µm of venule; and perfusion from 225.8 ± 26.81 PU to 140.8 ± 6.53 PU.

Analysis of plasma taken from rats chronically-treated with myrcene revealed that the terpene had no significant effect (*p* > 0.05) on circulating cytokine levels compared to vehicle-treated animals (Table 1).

### 2.4. Repeated Myrcene Treatment and Joint Histopathology

Twenty-one days following intra-articular injection of Freund’s complete adjuvant, rat knees showed moderate levels of joint damage including macrophage infiltration, pannus formation, cartilage damage and bone resorption (Figure 6). Multiple treatment of arthritic joints with myrcene had no effect on synovitis, pannus formation, cartilage destruction or bone resorption (Figure 6).

## 3. Discussion

*Cannabis* is a complex plant that contains various families of compounds including cannabinoids, flavonoids, and terpenes. While a growing body of evidence highlights the therapeutic benefits of cannabinoids, less is known about the physiological effects of non-cannabinoid molecules. It has been postulated that *Cannabis* constituents may interact with each other in the so-called “entourage effect” to either augment or diminish the efficacy of the individual components. This study aimed to assess the influence of myrcene on arthritis pain, tissue damage, and inflammation while the combination effects of myrcene and CBD were also investigated.

Anecdotally, arthritis patients prefer *Cannabis* strains that are rich in myrcene although the reason for this is unknown. Here, it was found that acute and repeated administration of myrcene to rats with chronically inflamed knees reduced evoked pain as measured by von Frey hair algesiometry. These data are consistent with other studies which showed that myrcene treatment could inhibit prostaglandin release and thermal hyperalgesia following acute inflammation [8]. Similarly, systemic administration of myrcene was able to reduce acetic acid-induced writhing and thermal hyperalgesia in naïve animals [7,9]. Several terpenes have been found to produce anxiolytic effects so it is feasible that myrcene could be acting on higher centers in the brain to alter the emotional component of chronic pain [32]. In the present study, however, myrcene was delivered locally to the joint so it is most likely that the terpene was causing analgesia by acting on peripheral nociceptors. Nevertheless, future experiments comparing the effect of myrcene on the activity of primary afferents, second-order spinal neurones, and affective regions of the brain such as the amygdala are required to elucidate the precise sites of action of the drug. The anti-nociceptive effect of myrcene was blocked by either a CB_1_ or a CB_2_-receptor antagonist, indicating the involvement of the endocannabinoid system. While previous studies suggest that myrcene stimulates the endogenous opioid system and inhibits transient potential vanilloid-1 (TRPV1) activity [7,9,33], this is the first indication that this terpene may also act via cannabinoid receptors to reduce pain. The analgesic property of myrcene is similar to another cannabis-derived terpene β-caryophyllene that also acts on cannabinoid and opioid receptors to reduce pain [7,34]. Peripheral myrcene had no effect on animal activity (data not shown), which corroborates findings elsewhere that indicated that myrcene at the doses tested does not produce somnolescence [9].

The present study also found that myrcene reduced joint inflammation in the adjuvant monoarthritic knee. In early stages of the disease, acute myrcene reduced leukocyte rolling, but in the chronic phase of the model with repeated drug administration, it also reduced cell adhesion and vasodilatation. The reduction in leukocyte kinetics is likely due to an effect of myrcene on vascular adhesion molecules. Cellular rolling is regulated by P-selectin and E-selectin while leukocyte adhesion to the vascular endothelium involves ICAM and VCAM molecules [35,36]. Thus, in day 7 arthritic knees myrcene may be inhibiting selectins only while at day 21 of the model myrcene could also be blocking the action of adhesion molecules. Previous work has shown that cannabis-related terpenes can downregulate the expression of selectins and adhesion molecules leading to reduced inflammation [37,38,39]. The identity of the adhesion molecules altered by myrcene requires further investigation. Although myrcene has been shown to inhibit cytokine production in the inflamed lung and kidney [40,41], circulating cytokine levels were not affected by local injection around the arthritic joint. The peripheral site of myrcene administration presented here may have limited its ability to reduce systemic cytokine levels.

Cannabis constituents have been shown to have a protective effect on joint damage in models of arthritis. Cannabidiol, for example, inhibited the progression of adjuvant-induced arthritis [42] and peripheral neuropathy in a model of osteoarthritis [22]. In vitro, repeated exposure of chondrocytes to myrcene reduced the expression of matrix metalloproteinases, which are known to cause joint destruction [10]. Here, chronic myrcene treatment had no effect on the clinical damage caused by adjuvant monoarthritis and this may be due to the severity of the model and/or inadequate dosing. Further studies are required to interrogate any potential protective effect of myrcene on the pathogenesis of inflammatory joint disease by increasing dose and mode of administration.

In light of the entourage phenomenon, this study assessed whether the combination of myrcene and CBD could have a greater effect on pain and inflammation compared to the terpene alone. Low doses of CBD and myrcene were chosen to allow for any potential synergism to be observed. When administered together, acute CBD and myrcene reduced joint pain and inflammation; however, no synergistic effect was detected. These results suggest that myrcene and CBD do not synergize in this model of arthritis.

### Study Limitations

The main limitation of this study was the low number of doses of myrcene and CBD used. The solubility restrictions of myrcene meant that there was a finite concentration that could be achieved without the drug coming out of solution. A low dose of CBD was tested so as to maximize any synergistic effect of the cannabinoid; however, further experiments using higher doses of CBD would have helped to optimize the analysis of entourage. A further limitation was the restriction of drugs locally to the joint to evaluate peripheral effects only. The rationale for this was to provide a proof-of-concept analysis of myrcene as a potential topical treatment. Future experiments administering myrcene systemically will provide a more global effect of the terpene in modulating joint pain and inflammation. Finally, using other antagonists, for example against TRPV1, would elucidate alternative targets that myrcene could be acting through in the joint.

## 4. Methods

Male Wistar rats (236–432 g) were purchased from Charles River Laboratories (Senneville, Quebec City, QC, Canada) and housed two animals per cage in ventilated racks at a temperature of 22 ± 2 °C with a 12 h light:12 h dark cycle. (lights on 7:00–lights off 19:00). Cages were lined with sterile wood chips and animals had free access to environmental enrichments, standard rat food and drinking water. All animals acclimated to the facility for one week prior to experimentation. All animal ethics protocols were approved by the Dalhousie University Committee on Laboratory Animals (UCLA), which adheres to the Canadian Council of Animal Care (CCAC) guidelines.

### 4.1. Induction of Adjuvant Monoarthritis

Deep anaesthesia was induced using isoflurane inhalation (2–5% in 100% O_2_; 1 L/min) until absence of the flexor withdrawal reflex to noxious toe pinch and the corneal blink reflex. The right stifle (knee) joint was shaved and swabbed with 70% ethanol (Fisher Scientific Company, Ottawa, ON, Canada). The knee was flexed and 50 µL of Freund’s complete adjuvant was injected into the joint cavity using a 30 G needle. The joint was then gently flexed and extended for ten seconds to help disperse the adjuvant throughout the intra-articular space. The knee was again swabbed with ethanol and the animals were allowed to recover under a heat lamp.

### 4.2. Pain Behavior Assessment

Secondary mechanical allodynia was determined by von Frey hair algesiometry based on the procedure of Chaplan et al. [43]. Briefly, animals were acclimated to a quiet testing room for at least one hour prior to pain assessment. Animals were placed in a Plexiglass chamber (31 cm × 9.5 cm × 25 cm) with a wire mesh floor. Calibrated von Frey hairs (North Coast Medical, Gilroy, CA, USA) were sequentially applied to the volar surface of the ipsilateral hindpaw and held for two seconds. A withdrawal response was signified by retraction of the paw and/or licking of the paw area. If no response was observed, a thicker filament was applied up to a cut-off force of 5.18 g.

The 50% withdrawal threshold was calculated using the following equation:$$50\%\;Threshold=\frac{\left(10^{\left[Xf+\kappa \delta \right])}\right)}{10,000}$$
where Xf is the bending force of the last von Frey filament used (in log units), κ is the tabular value for the pattern of the last six responses and δ is the mean difference (in log units) between stimuli.

### 4.3. Joint Inflammation Assessment

Articular leukocyte trafficking and blood flow were quantified by intravital microscopy and laser speckle contrast analysis respectively. Animals were deeply anesthetized with 25% urethane (2 mL loading dose; 0.5–0.7 mL top-up doses every 20 min) until cessation of the ocular blink and pedal withdrawal reflexes. Internal body temperature was measured by a rectally inserted thermometer and core temperature was maintained at 37 ± 1 °C by a thermostatically controlled heating pad (TC-1000, CWE Inc., Ardmore, PA, USA). The anesthetized animal was placed supine on a surgical board and a longitudinal midline incision was made in the neck. The sternohyoid muscle was blunt dissected to expose the trachea, which was subsequently cannulated to ensure a clear, unobstructed airway.

The left carotid artery was then isolated and catheterized using a fine cannula containing warmed, 1% heparinized saline. The carotid artery cannulation was connected to a BLPR2 pressure transducer and thence to a calibrated bridge amplifier (BP-1, World Precision Instruments, Sarasota, FL, USA) to allow continuous measurement of mean arterial pressure. The left jugular vein was then isolated and cannulated with tubing also containing warmed 1% heparinized saline.

Finally, a small ellipse of skin was excised from over the ipsilateral knee joint and the underlying fascia removed to expose the joint microvasculature, which was kept moist by perfusion with 0.9% saline. For intravital microscopy, animals first received an i.v. injection of Rhodamine 6 G (0.1 mL; 5 mg/10 mL; Sigma Aldrich, St. Louis, Missouri, USA) to label circulating leukocytes. A Leica DM2500 microscope, (Leica Microsystems Canada, Inc., Richmond Hill, ON, Canada) with a HCX APOL 20X objective and HC Plan 10X eyepiece was used to observe capsular leukocyte trafficking under a 530 nm fluorescent light. Straight, unbranched post capillary venules (15–50 μm) located in the knee joint were selected and three, one-minute videos were recorded at a final magnification of 200× using a Leica DFC 3000 camera (Leica Microsystems Canada Inc., Richmond Hill, ON, Canada). The videos were then analyzed offline in a blinded manner, where rolling and adherent leukocytes were quantified. Rolling leukocytes were defined as cells moving slower than blood flowing past an arbitrary line perpendicular to the vessel of interest. Leukocytes were considered adherent if they remained immobile for more than 30 s within a 100 μm span of the postcapillary venule. The number of rolling and adherent leukocytes was quantified for each of the one-minute recordings.

Joint perfusion was carried out by laser speckle contrast analysis (LASCA) using the PeriCam PSI System (Perimed Inc., Ardmore, PA, USA). A one-minute recording was taken at a working distance of 10 cm, and a frame capture rate of 25 images/second using PIMSoft software (Version 1.5.4.8078; Perimed Inc., Ardmore, PA, USA). Post euthanasia, a “dead scan” recording was taken of the joint and subtracted from experimental perfusion values as a biological zero to account for any cellular Brownian motion or tissue optical interference. The speckle pattern was subsequently analyzed offline in a blinded manner and recorded in arbitrary perfusion units (PU) for each time-specific recording. Blood perfusion corresponding to the joint capsule was recorded.

### 4.4. Cytokine Analysis

Whole blood samples were collected by intracardiac puncture using a 5 mL syringe at endpoint on day 21. The sample was transferred immediately into a 6 mL EDTA-coated Eppendorf tube and placed on ice. Samples were centrifuged at 1000× *g* at 4 °C for 10 min. Plasma supernatant was then transferred into Eppendorf tubes and stored at −80 °C until use. Frozen aliquots were thawed on ice and centrifuged at 10,000× *g* for 5–10 min, and then diluted with 1X universal assay buffer. Antigen standards and a 4-fold serial dilution of the reconstituted standard were prepared as per the ProcartaPlex™ Mutliplex Immunoassay for Convenience and the Mix & Match Panels User Guide. The magnetic beads were vortexed for 30 s and 40 μL of the beads was added to each well, before subsequently removing the liquid and adding 150 μL of wash buffer. Standards and pre-diluted plasma samples (50 μL) were added to the bead-coated wells. Blank wells were filled with 50 μL of universal assay buffer. The plate was subsequently incubated at room temperature on a microplate shaker for 2 h. Post-incubation, the plate was washed a total of three times. Twenty microliters of the detection antibody mixture were added to each well and incubated at room temperature for 30 min. Next, 40 μL of Streptavidin-PE (SAPE) was added to each well and incubated at room temperature while shaking for a further 30 min. The plate was then prepared for analysis by adding 120 μL of Reading Buffer into each well and incubating for five minutes. A Bio-Plex 200 system (Bio-Rad, Hercules, CA, USA) was used to quantify specific cytokines (interleukin-1β (IL-1β), IL-10, IL-17A, IL-6 and tumor necrosis factor-α (TNF-α)).

### 4.5. Joint Histopathology

At day 21 after FCA injection, four myrcene-treated and four vehicle-treated knees from the chronic experiments (see below) were assessed for joint pathology. Deeply anesthetized animals were perfused trans-cardially with saline, followed by 4% paraformaldehyde. The joint was isolated and immersed in 4% paraformaldehyde for 24 h before being washed with sterile water and transferred to 70% ethanol for storage. Joint histopathology was assessed by Bolder BioPATH Inc. (Boulder, CO, USA). Knee joint were initially decalcified for 4–5 days in 5% formic acid and then halved in the frontal plane and embedded in mounting paraffin. Joints were subsequently sectioned (8 µm sections) at 200 µm intervals across the whole knee. Sections were stained with 0.04% toluidine blue and assessed microscopically for signs of disease. The joint sections were then scored for four parameters (synovitis, pannus formation, cartilage damage, and bone resorption) using the previously described OARSI scoring system [44]. Each parameter was scored 0–5 to give a possible total summed score out of 20.

### 4.6. Experimental Treatment Protocols

The acute effects of myrcene on adjuvant monoarthritic pain and inflammation were initially assessed on day 7 of the model. von Frey hair algesiometry was measured before (baseline, BL) and then over 3 h following administration of either vehicle (soybean oil, 50 µL s.c. over knee joint) or myrcene (1 and 5 mg/kg, 50 µL s.c. over knee joint). Dose was based on a systemic dose used in a previous study [7]. To examine cannabinoid receptor involvement in myrcene responses, separate cohorts of rats were pre-treated with either the CB_1_ receptor antagonist AM281 or the CB_2_ receptor antagonist AM630 (75 µg in 50 µL s.c. over the knee 10 min prior to 1 mg/kg myrcene injection). For joint inflammation experiments, a similar protocol was employed in separate groups of animals where intravital microscopy and LASCA were carried out over a 60 min period. The effect of myrcene on hindlimb mechanosensitivity and inflammation were also investigated in naïve animals as a further control.

To explore any synergistic effects between the phytocannabinoid CBD and myrcene, a combination regimen was employed. In day 7 adjuvant monoarthritic rats, low-dose CBD (200 µg) was co-administered with low-dose myrcene (1 mg/kg) in a single 50 µL bolus injected s.c. over the knee. This dose of CBD had no effect on joint inflammation or pain (data not shown). Following treatment, von Frey hair algesiometry was measured over 3 h while in separate animals intravital microscopy and LASCA were tested over 60 min.

For chronic studies, myrcene (5 mg/kg, 50 µL s.c. over knee joint) was administered on days 1, 2, 3, 7, 10, 14 and 21 after monoarthritis induction. Secondary allodynia was measured before (baseline: BL) and then prior to each injection of myrcene. Leukocyte trafficking and synovial blood flow were then tested in the same animals at day 21 after Freund’s adjuvant injection. Blood samples were then taken for cytokine analysis and ipsilateral knee joints harvested for histopathological assessment.

### 4.7. Reagents

Myrcene (7-methyl-3-methylene-1,6-octadiene) in soy oil was purchased from Toronto Research Chemicals (Toronto, ON, Canada). CBD (2-[(1R,6R)-3-methyl-6-(1-methylethenyl)-2-cyclohexen-1-yl]-5-pentyl-1,3-benzenediol) was obtained from Tocris Bioscience (Bio-Techne, Abingdon, UK). AM281 (CB1 receptor antagonist; 1-(2,4-dichlorophenyl)-5-(4-iodophenyl)-4-methyl-N-4-morpholinyl-1H-pyrazole-3-carboxamide) and AM630 (CB2 receptor antagonist; 6-iodo-2-methyl-1-(2-morpholin-4-ylethyl)indol-3-yl]-(4-methoxyphenyl)methanone) were obtained from Cayman Chemicals (Ann Arbor, MI, USA). The selectivity of AM281 is 12 nM: 4200 nM for CB_1_:CB_2_, and for AM630 5µM: 31 nM for CB_1_:CB_2_. Rhodamine 6 G, cremophor, dimethyl sulphoxide (DMSO), urethane, and Freund’s complete adjuvant were obtained from Sigma Aldrich (St. Louis, MO, USA). Solutions of CBD, AM281, and AM630 were prepared in vehicle (1:1:18; DMSO:cremophor:saline) on the day of use. Rhodamine 6 G (0.05%) was dissolved in 0.9% saline.

### 4.8. Statistics

All data were presented as an average of the percent baseline ± SEM. Normality was determined using the Kolmogorov–Smirnov test and compliant data were analyzed by parametric statistics, i.e., one- or two-factor repeated measures analysis of variance (RMANOVA) with Bonferroni’s post hoc multiple comparisons test, or Student’s t-test. Joint histopathology scores and cytokine concentrations were analyzed by the non-parametric Mann–Whitney U test. A probability value of *p* < 0.05 was considered statistically significant.

## 5. Conclusions and Future Directions

In summary, myrcene was found to have anti-inflammatory and analgesic effects in inflammatory joint disease by activating articular cannabinoid receptors. In vitro studies showing myrcene signaling via cannabinoid receptors; however, this still needs to be tested. While chronic myrcene treatment had no effect on joint pathology, long-term administration of the compound had a more profound effect on inflammatory parameters. Peripheral myrcene had no effect on circulating cytokine levels so the anti-inflammatory mechanism of action still needs to be resolved. Finally, at the doses tested, co-administration of myrcene with CBD failed to produce any synergistic response, suggesting a lack of entourage effect between these two compounds. Nevertheless, future studies should still explore possible interactions between *Cannabis*-derived terpenes and cannabinoids in the control of joint pain and inflammation. Together, these findings may explain why arthritis patients prefer *Cannabis* strains rich in myrcene to help manage their pain and inflammation.

## Figures and Tables

**Figure 1 ijms-23-07891-f001:**
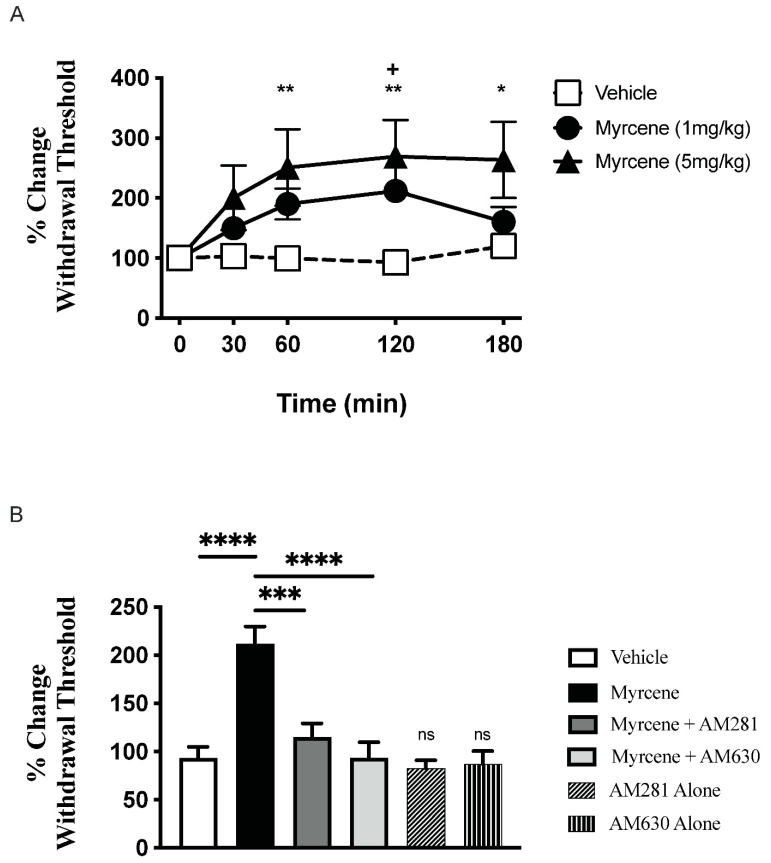
Acute administration of myrcene dose-dependently reduced secondary allodynia in adjuvant monoarthritic knees. Local injection of myrcene around the arthritic knee reduced von Frey hair withdrawal threshold at day 7 of the model with the maximal response occurring 120 min after drug administration (**A**). ** *p* < 0.01, * *p* < 0.05 (5 mg/kg dose) ^+^ *p* < 0.05 (1 mg/kg dose) two-factor RMANOVA with Bonferroni’s post hoc test. Pre-treatment of arthritic animals with either the CB_1_-receptor antagonist AM281 (75 µg s.c.) or the CB_2_-receptor antagonist AM630 (75 µg s.c.) blocked the analgesic effect of myrcene (**B**). Antagonists alone were not significantly different from vehicle. ns—not significant, **** *p* < 0.0001, *** *p* < 0.001, one-factor RMANOVA with Bonferroni’s post hoc test. Data are means ± S.E.M.

**Figure 2 ijms-23-07891-f002:**
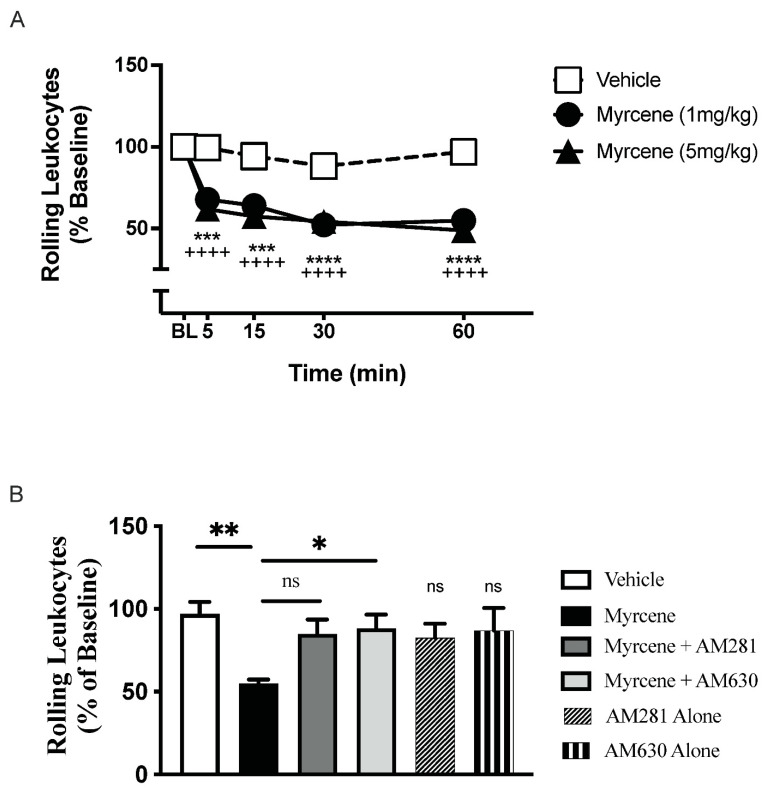
Acute myrcene treatment altered the kinetics of leukocytes in the arthritic joint microvasculature. Topical administration of myrcene to day 7 adjuvant monoarthritic knees reduced the number of rolling leukocytes in post-capillary venules supplying the joint (**A**). The maximal response occurred at 60 min following the start of treatment. **** *p* < 0.0001, *** *p* < 0.001 (5 mg/kg dose) ^++++^ *p* < 0.0001 (1 mg/kg dose) two-factor RMANOVA with Bonferroni’s post hoc test. Compared to myrcene alone, the reduction in leukocyte rolling was blocked by the CB_2_-receptor antagonist AM630 (75 µg s.c.) but not the CB_1_-receptor antagonist AM281 (75 µg s.c.). (**B**). Antagonists alone were not significantly different from vehicle. ns—not significant, ** *p* < 0.01, * *p* < 0.05, one-factor RMANOVA with Bonferroni’s post hoc test. Data are means ± S.E.M.

**Figure 3 ijms-23-07891-f003:**
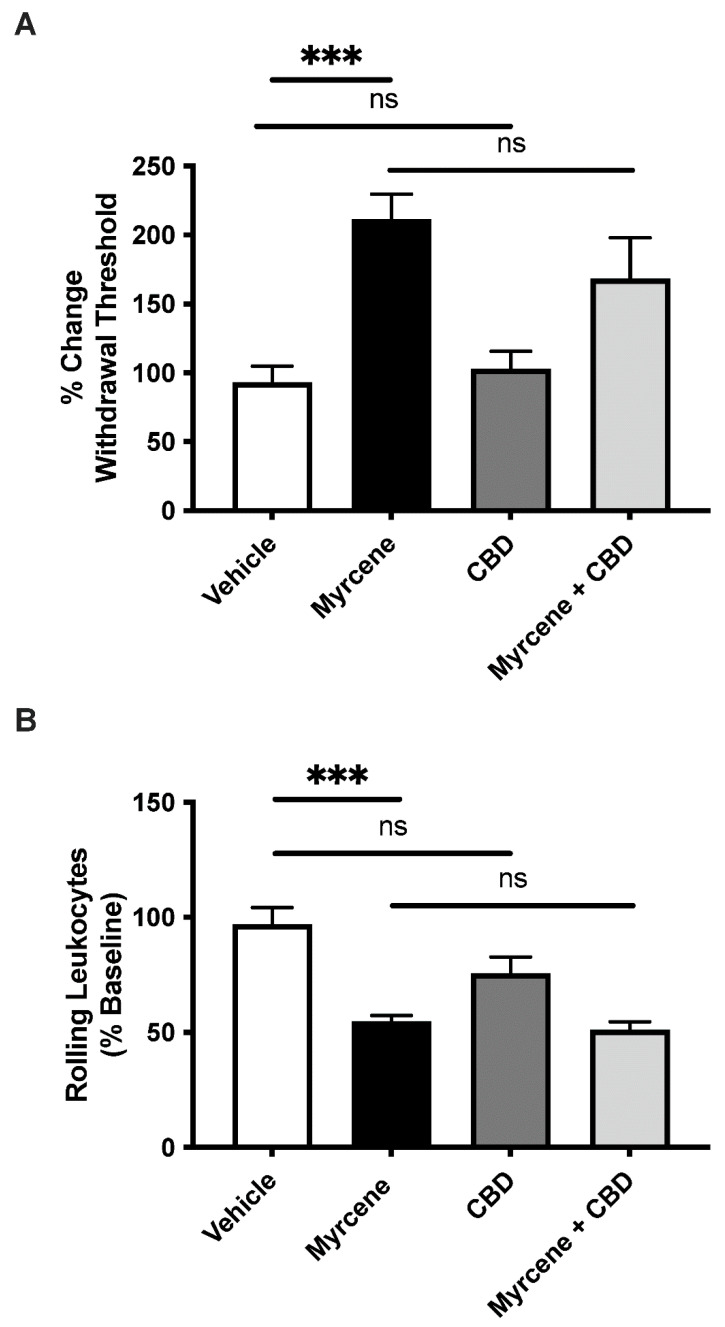
The effect of co-administration of low-dose cannabidiol (CBD) and myrcene on joint pain and inflammation. While low-dose myrcene reduced hindlimb pain (**A**) and leukocyte rolling (**B**), the addition of cannabidiol (CBD) had no additional effect on these analgesic or anti-inflammatory responses. ns—not significant, *** *p* < 0.001, one-factor RMANOVA with Bonferroni’s post hoc test. Data are means ± S.E.M.

**Figure 4 ijms-23-07891-f004:**
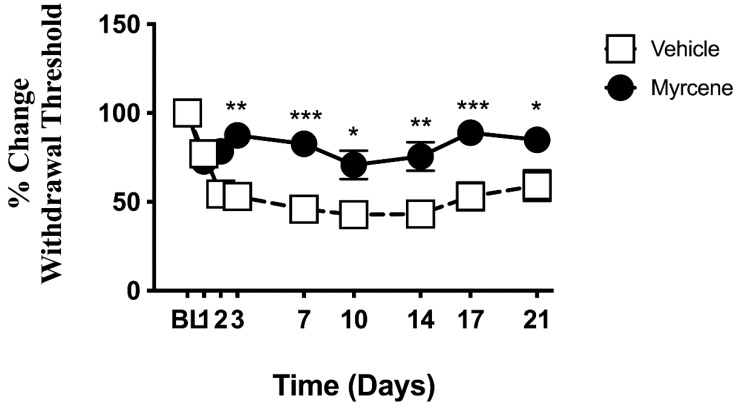
The effect of repeated myrcene treatment on chronic knee joint pain. Local administration of myrcene on days 1, 2, 3, 7, 10, 14 and 21 after monoarthritis induction reduced chronic joint pain across the time course. Pain behavior was tested before myrcene injection at each time point to negate acute responses to the drug. *** *p* < 0.001, ** *p* < 0.01, * *p* < 0.05 two-factor RMANOVA with Bonferroni’s post hoc test. Data are means ± S.E.M.

**Figure 5 ijms-23-07891-f005:**
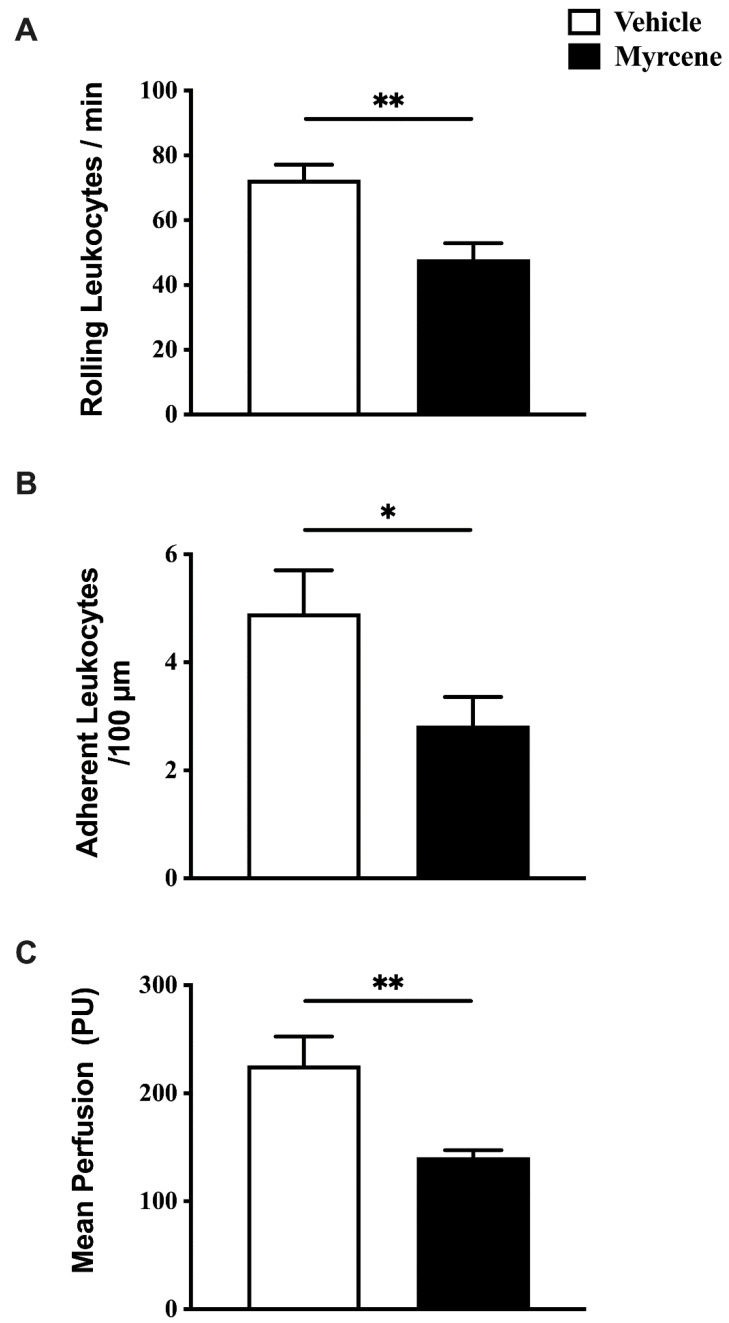
Inflammatory changes following chronic administration of myrcene to adjuvant monoarthritic knees. Repeated, local injection of myrcene decreased articular leukocyte rolling (**A**), adherence (**B**), and blood flow (**C**), at day 21 after monoarthritis induction. ** *p* < 0.01, * *p* < 0.05 Student’s *t*-test. Data are means ± S.E.M.

**Figure 6 ijms-23-07891-f006:**
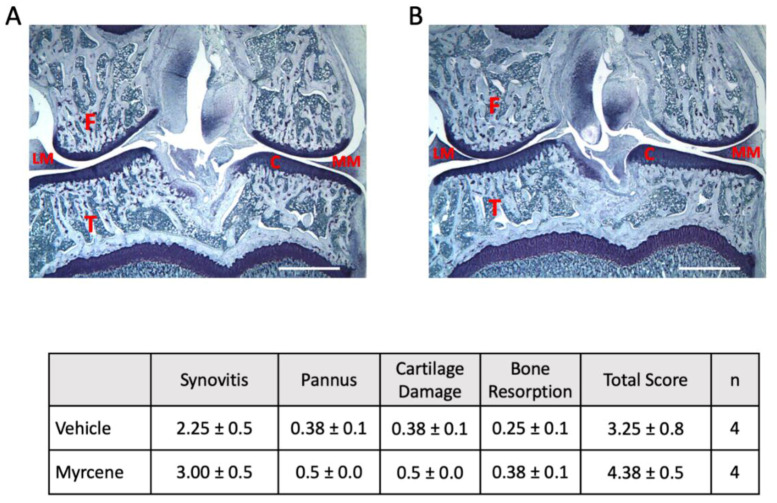
Lack of effect of chronic myrcene treatment on joint histopathology. Representative histological sections of day 21 adjuvant monoarthritic knees (frontal plane) following repeated administration of either vehicle (**A**) or myrcene (**B**). Table shows that myrcene had no effect on the joint damage scores across the four parameters. C: joint cartilage; F: femur; LM: lateral meniscus; MM: medial meniscus; T: tibia. Scale bar = 2.5 mm.

**Table 1 ijms-23-07891-t001:** Plasma cytokine levels in vehicle compared to myrcene-treated animals. The concentration of inflammatory cytokines (pg/mL) was unaltered by chronic myrcene treatment. Data presented as mean concentration ± S.E.M.

	IL-1β	IL-10	IL-6	IL-17A	TNF-α	*n*
Vehicle	631 ± 258	277 ± 83	80 ±32	41 ± 16	84 ± 46	7
Myrcene	631 ± 204	263 ± 67	80 ± 24	40 ± 12	96 ± 36	7–8

## Data Availability

Data are available from the corresponding author upon request.

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
