# Peer review of "Anti-Inflammatory and Analgesic Properties of the Cannabis Terpene Myrcene in Rat Adjuvant Monoarthritis"

_ijms, 2022, doi:10.3390/ijms23147891_

Round 1

Reviewer 1 Report

The investigation has been well designed and has been clearly describe in all the sections of the manuscript. Figures are clear and easily readable.  The authors show confidence both with the treated topic and with the experimental techniques adopted. I recommend the publication of the manuscript in the present form. 

Author Response

We are very grateful to the reviewer for their supportive assessment of our work.

Reviewer 2 Report

The objective of this study was to characterize the effect of myrcene on joint pain and inflammation using a model of chronic arthritis.

After carefully reading the manuscript, I conclude that the abstract, introduction, and other chapters cover the issues discussed in an extensive and proper manner.

Although it is a valuable work having an interesting idea it needs some adjustment:

  •             It is quite surprising that there is no concise description of the conclusion and future perspective. The authors should make section covering the future perspective and conclusion in the form of an additional chapter.
  •             The authors also should add section of a concise description of the limitations of the study in the form of an additional chapter.
  • The small number of citations of works from the last 5 years is rather surprising (out of 31 bibliographic items, only 11 works have been published after 2017). I believe that the topic has been developing vigorously over the past 5 years and focusing on old articles is not the best approach to reviewing it. Therefore, I am convinced that the authors must carry out a detailed literature review and supplement the citations with references to the latest research in this field.

I recommend publication after minor revision.

Author Response

After carefully reading the manuscript, I conclude that the abstract, introduction, and other chapters cover the issues discussed in an extensive and proper manner. 

We kindly thank the reviewer for their excellent suggestions to improve our manuscript. Our replies appear below.

Although it is a valuable work having an interesting idea it needs some adjustment:

  •             It is quite surprising that there is no concise description of the conclusion and future perspective. The authors should make section covering the future perspective and conclusion in the form of an additional chapter.

REPLY: We have added a section entitled “Conclusions and Future Directions” where we summarise our work (p. 27).

  •             The authors also should add section of a concise description of the limitations of the study in the form of an additional chapter.

REPLY: A “Study Limitations” section has been added to the manuscript in addition to some limitations appearing throughout the Discussion (p. 26).

  • The small number of citations of works from the last 5 years is rather surprising (out of 31 bibliographic items, only 11 works have been published after 2017). I believe that the topic has been developing vigorously over the past 5 years and focusing on old articles is not the best approach to reviewing it. Therefore, I am convinced that the authors must carry out a detailed literature review and supplement the citations with references to the latest research in this field.

REPLY: We have added an additional 13 references to the manuscript which include more recent reports of the subject area.

I recommend publication after minor revision.

Reviewer 3 Report

This is an interesting study that investigated the effects of cannabis terpene myrcene on joint pain and inflammation. Results indicated that local application of myrcene to adjuvant monoarthritic knees reduced secondary allodynia. CB1-receptor blockade blocked the analgesic effect of myrcene. Myrcene also reduced leukocyte infiltration. The results indicate that these effects were not synergized by cannabidiol. Chronic administration of myrcene reduced leukocyte adherence and synovial blood flow but did not affect cytokine levels. Although the study addresses the important subject of identification of novel non opioid approaches to pain management, provides important results and depends on pharmacology as well as behavioral tests, it lacks cutting-edge tools such as mouse genetics and in vivo activity imaging.

Comments

11-    It is essential to discuss CB receptor expression. Mention the type of neurons and tissues that express CB1 and CB2 receptors, its regulation and role in mediating analgesia.

22-    Discuss expression of CB receptors in leukocytes.

33-    It would be also great to add CB receptor activation data by myrcene.

44-    Results section 1 and figures 1A, 1B, 2A and 2B should include data on CB receptor blockers alone. The study also lacks control data using specific CB receptor agonists. Such data is essential to understand the mechanism of action of myrcene in the absence of knockout mouse models.

55-    Discuss the affinity and specificity of CB1 and CB2 blockers used.

66-    It is clear that myrcene has an inhibitory effect on the sensory component of pain, please discuss its effect on emotional-motivational component of pain and spontaneous pain.

77-    Does myrcene have an effect on leukocyte infiltration into brain neurons such as in the amygdala, PFC, PAG and other regions known to be important in pain and anxiety?

88-    Discuss the limitations of the study.

Author Response

We appreciate the Reviewer’s comments and their suggestions have certainly strengthened the manuscript. Thank you.

Comments

11-    It is essential to discuss CB receptor expression. Mention the type of neurons and tissues that express CB1 and CB2 receptors, its regulation and role in mediating analgesia.

REPLY: A paragraph has been added to the Introduction outlining the tissue expression of cannabinoid receptors and their physiological effects in joints. The mechanism by which cannabinoids impart analgesia has also been included (p. 4).

22-    Discuss expression of CB receptors in leukocytes.

REPLY: The expression of cannabinoid receptors on leukocytes is now described in the Introduction (p. 4).

33-    It would be also great to add CB receptor activation data by myrcene.

REPLY: We agree that in vitro  experiments examining the direct activation of cannabinoid receptors by myrcene would be useful and highlight this as potential future direction (p. 27).

44-    Results section 1 and figures 1A, 1B, 2A and 2B should include data on CB receptor blockers alone. The study also lacks control data using specific CB receptor agonists. Such data is essential to understand the mechanism of action of myrcene in the absence of knockout mouse models.

REPLY: We now include new data showing that the cannabinoid antagonists alone have no effect on pain behaviour or leukocyte trafficking in these studies (please see new Figures 1B and 2B).

We have previously published the effect of selective CB1 and CB2 receptor agonists on joint vascular reactivity and pain in rats (Schuelert, N.; McDougall, J.J. doi:10.1002/art.23156; Schuelert, N. et al. doi:10.1016/j.joca.2010.09.005; Krustev et al. doi:10.1186/s13075-014-0437-9; McDougall et al. 153, 358-366, doi:0707565 [pii]). Reference to this body of work has been added to the paper (p.4).

55-    Discuss the affinity and specificity of CB1 and CB2 blockers used.

REPLY: the selectivity of the antagonists has been added to the Methods (p. 12).

66-    It is clear that myrcene has an inhibitory effect on the sensory component of pain, please discuss its effect on emotional-motivational component of pain and spontaneous pain.

REPLY: Indeed there is evidence that some terpenes can have affective properties and could alter the emotional aspect of the pain experience. We don’t think higher centres are being activated in our study due to the local application of the drugs; however, we now highlight this as a testable hypothesis in the Discussion (p. 24).

77-    Does myrcene have an effect on leukocyte infiltration into brain neurons such as in the amygdala, PFC, PAG and other regions known to be important in pain and anxiety?

REPLY: We are not aware of any data showing myrcene having an effect on leukocyte recruitment in the brain although this is a very interesting idea that should be explored by others.

88-    Discuss the limitations of the study.

REPLY: A “Study Limitations” section has been added to the Discussion (p. 26). Other limitations of the study appear throughout the Discussion.

Round 2

Reviewer 2 Report

I thank the authors for their cooperation. I believe that the revised manuscript has improved in clarity. I wish the authors a large group of manuscript readers.

Reviewer 3 Report

The authors fully addressed my comments.